# The Seminal Role of the Proinflammatory Cytokine IL-1β and Its Signaling Cascade in Glioblastoma Pathogenesis and the Therapeutic Effect of Interleukin-1β Receptor Antagonist (IL-1RA) and Tolcapone [note 1]

**DOI:** 10.3390/ijms26146893

**Published:** 2025-07-18

**Authors:** Jagadeesh Narasimhappagari, Ling Liu, Meenakshisundaram Balasubramaniam, Srinivas Ayyadevara, Orwa Aboud, W. Sue T. Griffin

**Affiliations:** 1Donald W. Reynolds Department of Geriatrics, Reynolds Institute on Aging, University of Arkansas for Medical Sciences, Little Rock, AR 72205, USA; njagadeesh@uams.edu (J.N.); liuling@uams.edu (L.L.); mbalasubramaniam@uams.edu (M.B.);; 2Central Arkansas Veterans Healthcare System, Little Rock, AR 72205, USA; 3Department of Neurology and Neurological Surgery, University of California, Davis, CA 95616, USA; oaboud@ucdavis.edu

**Keywords:** glioblastoma, IL-1β, IL-1RA, Tolcapone, MyD88, NF-κB, autophagy, apoptosis

## Abstract

Interleukin-1 beta(IL-1β) is the major driving force in neuroinflammation. Here, we report on (i) the role of (IL-1β) in activating a signaling cascade that leads to proliferation and metastasis in glioblastoma cancer pathogenesis as well as (ii) the therapeutic role for IL-1 Receptor Antagonist (IL-1RA) and Tolcapone against untoward aspects of tumor pathogenesis. Here, we report that IL-1β treatment at 50 ng/mL for 48 h increased proliferation and metastasis by 30-fold (*p* ≤ 0.05), leading to the formation of clones of rapidly dividing cancer cells, leading to the formation of organized glial fibrillary acid protein (GFAP)-immunoreactive, clone-like structures with protruding spikes. Further, IL-1β treatment significantly increased the expression of mRNA levels of the IL-1β-driven pathway TLR-MyD88-NF-κB-TNFα and IL-6 (*p* ≤ 0.05). IL-1β also increased autophagy via elevation of mRNA and protein levels of cathepsin B, LAMP-2, and LC3B. In contrast, IL-1RA and Tolcapone inhibited this proliferation and the expression of these mRNAs and proteins, inhibiting autophagy by downregulating these autophagy proteins and inducing apoptosis by upregulating the expression of pro-apoptotic proteins like caspase-8 and caspase-3. IL-1β and its receptor can be targeted for successful anticancer therapy, as shown here with the use of IL-1RA and/or Tolcapone.

## 1. Introduction

Neuroinflammation has long been associated as a causative factor in a variety of age-related pathologies, including Alzheimer’s disease, cardiovascular diseases, and type-2 diabetes, as well as brain cancers [1]. In fact, inflammation is a hallmark of cancer and is widely recognized to influence all stages of cancer pathogenesis, from normal cell transformation to metastatic cancer. In general, inflammation is the normal mechanism of the immune system to protect the organism or body from invading pathogens, which further activate inflammatory responses by releasing inflammatory cytokines, in particular IL-1β, to protect the body from invading-pathogen damage [2]. Inflammatory cytokines play important roles in the maintenance of cellular homeostasis, as they control a variety of important biological mechanisms, including the cell cycle, autophagy, and apoptosis [3]. Conversely, IL-1β can lead to activation of multiple signaling cascades that are known to have the potential to give rise to the development of Alzheimer’s disease, Parkinson’s disease, multiple sclerosis, and epilepsy [4,5,6]. Further, IL-1β has been reported to play a pivotal role in the growth of primary tumors and in the promotion of epithelial-to-mesenchymal transition (EMT), leading to migration and invasion in breast cancer [7]. In addition, IL-1β is known to regulate the expression of Nuclear Factor-κB (NF-κB), which plays an important role in cancer metastasis and progression [8] and is known to regulate the expression of COX-2, the inflammatory regulator of IL-6, which in turn is responsible for cancer metastasis via increased production of intracellular adhesion molecules [ICAMs and VCAMs] [9]. NF-κB also regulates the expression of members of anti-apoptotic proteins, including Bcl-2, Bcl-xL, and A1/Bfl-1, each of which is required for cancer cell survival in metastatic conditions [10], allowing cancer cells to adhere to the extracellular matrix, aiding metastasis and further organismal invasion. The expression of IL-6 also increases angiogenesis, an important hallmark of cancer; it is this formation of new blood vessels which provides the nutrients necessary for further tumor growth.

Calcineurin, which is regulated by the NF-κB pathway in cancer cells, controls cell cycle progression during the G1 phase, resulting in elevated proliferation for cancer cell growth [11,12]. Elevated levels of calcineurin and mutations in the gene PPP3CC are common occurrences in many cancers, resulting in elevated anti-apoptotic factors that directly enhance the transcriptional and translational events required for cancer metastasis. Taken together, these studies demonstrate that drugs targeting the NF-κB pathway can be efficacious in controlling glioblastoma pathology.

Hurmath et al., 2014, reported that overexpression of IL-1β in human glioblastoma cells promoted the proliferation and migration of U87MG and U251MG cells, an event that was blocked by treating the cells with IL-1Receptor antagonist (IL-1RA) [13], resulting in inhibition of migration, invasion, and proliferative effects, indicating that the effects were mediated by IL-1β and its receptor. IL-1 β significantly increased proliferation of U87MG glioma cells compared to control and interestingly there was no proliferation in U251MG glioma cells compared to control. The U87 cell line is used to study cancer growth and development, underlying molecular mechanisms, signaling pathways, and the tumor microenvironment. U-87MG cell lines are well-studied glioblastoma cell lines for drug screening and testing purposes, allowing researchers to identify new potential anticancer drugs and assess their efficacy and toxicity. This prompted us to use the U87MG cell line as a model cell line to understand the signaling mechanisms and to explore the therapeutic applications of IL-1RA and the FDA approved drug Tolcapone.

Tolcapone, a drug commonly used in the treatment of Parkinson’s disease, is a potent inhibitor of Catechol-O-methyltransferase (COMT) and is permeable across the blood–brain barrier [14]. Previous studies indicate that Tolcapone increases the bioavailability of dopamine in cells. Tolcapone is FDA approved in adult patients for the treatment of Parkinson’s disease as an adjunct therapy with levodopa, which is a dopamine precursor [15]. Interestingly, Tolcapone has shown promising anticancer potential with neuroblastoma (NB) cells in preclinical models by inhibiting COMT. Treating four established NB cells lines (SMS-KCNR, SH-SY5Y, BE (2)-C, CHLA-90) and two primary NB cell lines with Tolcapone for 48 h decreased cell viability in a dose-dependent manner and induced cell death by activating caspase-3-mediated apoptosis in neuroblastoma cell lines [16]. This prompted us to study the anticancer potential of Tolcapone against glioblastoma pathogenesis using U87 MG as a model cell line.

Previously we published an abstract reporting the role of the proinflammatory cytokine IL-1β and its signaling cascade in glioblastoma pathogenesis and the therapeutic effect of IL-1RA and Tolcapone as anticancer agents in Proceedings of the American Association for Cancer Research Annual Meeting 2025 [17]. This paper is an extended version of our abstract and here we delineated the IL-1β-driven signaling pathway that activates the MyD88-NF-κB and its downstream signaling cascade in U87 cells, leading to proliferation and metastasis. Treatment with IL-1RA inhibited these effects, providing evidence that these events are solely mediated by IL-1β via the IL-1 receptor. We also report for the first time that the FDA-approved drug Tolcapone, an enzyme catechol-O-methyltransferase inhibitor, is similarly effective in inhibiting proliferation and metastasis of glioblastoma.

IL-1RA and Tolcapone both inhibited the growth and metastasis of U87MG cancer cells via their inhibition of the NF-κB pathway. Furthermore, IL-1RA and Tolcapone treatments inhibited autophagy by downregulating autophagic proteins LC3B, LAMP-2, as well as Cathepsin B at both transcriptional and translational levels. In addition, IL-1RA and Tolcapone induced apoptosis via increasing the levels of apoptotic markers like Caspase-8 and Caspase-3, suggesting that both IL-1RA and Tolcapone are promising drugs against glioblastoma.

## 2. Results

### 2.1. IL-1β Induced Morphological Changes in Human Glioblastoma Cancer Cells Following Treatment with Either IL-1RA or Tolcapone

Elevated proliferation of U87 glioblastoma cells was noted 24 h after exposure to 50 ng/mL of IL-1β (Figure 1A,B). Forty-eight hours post treatment, the cells were clumped together clone-like, with elongated stick-like protuberances. Treatment with either IL-1RA or Tolcapone resulted in inhibition of cancer cell proliferation as well as notable decreases in clone-like formations, with IL-1RA noticeably more effective than Tolcapone.

### 2.2. IL-1β Treatment Elevates GFAP Expression in Human Glioblastoma Cancer Cells

Glial Fibrillary Acidic Protein in astrocytes is important in maintaining cell structure, cell movement, and communication. Immunocytochemical analysis showed that U87 cells are highly immunoreactive for GFAP, and 24 hr post treatment with IL-1β resulted in a 1.4-fold increase in the mRNA and protein levels of GFAP; an increase which was reduced by either IL-1RA or Tolcapone to 0.7-fold [*p*≤ 0.005] and 0.6-fold [*p*≤ 0.0005], respectively. Treatment of U87 cells with IL-1β elevated GFAP both mRNA and protein levels (Figure 2A,B), consistent with the idea that GFAP, because of its highly structural complexity noted here, guidance role, and cellular lysosomal autophagy functions, acts as an important factor in the known aggressive metastatic nature of this cancer within diverse regions of the brain.

### 2.3. Cytotoxic Effects of IL-1RA and Tolcapone on Human Glioblastoma Cancer Cells

To evaluate the effect of Tolcapone on U87 cell growth, the cells were treated for 48 h at varying concentrations [1–50 µM/mL], and cell viability was measured by MTT assay. As shown in Figure 2C, cells treated with 50 ng/mL of IL-1β had a 34.4% increase in proliferation compared to untreated cells. Tolcapone at 20 µM inhibited 50% of U87 cell growth, and IL-1RA at 1 µg/mL resulted in a 51% decrease in the IL-1β-induced cancer cell growth in human glioblastoma cancer.

### 2.4. IL-1RA and Tolcapone Inhibited the Formation of U87 Colonies and Cancer Invasiveness

The clonogenic assay was performed in IL-1β treated U87 cells to determine anchorage-dependent growth and reproductive cell division. As depicted in Figure 2D, both the size and number of U87 tumor spheres were increased significantly in IL-1β treated plates. Conversely, IL-1RA+IL-1β and Tolcapone reduced this clonogenicity compared to untreated control cells. These results demonstrate the profound clonogenic inhibitory potential of both IL-1RA and Tolcapone on U87 cells in vitro. In vitro cell motility and invasion scratch wound assays were performed to ascertain the functional implication of IL-1RA and Tolcapone-induced morphological changes in glioblastoma cancer cells. The rate of cell migration toward the scratch wound was significantly increased in IL-1β treated cells compared to that of untreated control cells (Figure 3). Both IL-1RA and Tolcapone [IC_50_] reduced this metastasis and the migratory effect of U87 cancer cells in a time-dependent manner, revealing anti-metastatic potential of IL-1RA and Tolcapone.

### 2.5. IL-1β Increased the Transcription and Translation of the MyD88-NFĸB-IL-6-Calcineurin Pathway

Treatment of U87 cells with IL-1β at 50 ng/mL increased expression levels of both the mRNAs and proteins of the TLR2-MyD88-NFĸB-IL-6-calcineurin pathway. These increased levels themselves can be correlated with the known highly metastatic nature of U87 glioblastoma in the brain. IL-1β treatment significantly increased the expression of mRNA levels of MyD88 [*p* ≤ 0.01], NFĸB [*p* ≤ 0.02], IL-6 [*p* ≤ 0.002], and calcineurin [*p* ≤ 0.01] (Figure 4A). IL-1β treatment also increased the protein levels of MyD88-NFĸB-IL-6-calcineurin (Figure 4B) but not of TLR in U87 cells. Further, IL-1β at 50 ng/mL significantly increased the expression of the protein levels of MyD88 [*p* ≤0.01], NFĸB [*p* ≤0.02], IL-6 [*p* ≤0.002], and PPP3 [*p* ≤ 0.01] (Figure 4C). Treatment with IL-1RA decreased this effect in a dose-dependent manner. Tolcapone at 20 µM reduced NFĸB-IL-6-Calcineurin but did not show any effect on MyD88. IL-1RA at 1 µg/mL and Tolcapone at 20µM significantly decreased the expression of the NFĸB-IL-6-Calcineurin protein levels.

### 2.6. IL-1β Increased the Transcription and Translation of Components of Autophagy

Autophagy is a cellular process that clears cells of metabolic waste, and in cancer cells autophagy plays a beneficial role in cancer development and progression, favoring cancer proliferation and metastasis. Autophagy modifies the tumor microenvironment along with promotion of angiogenesis.

IL-1β treatment of U87 cells significantly increased the mRNA and protein levels of two of the three necessary autophagy proteins, viz., LC3B and LAMP2 (Figure 5A–C). IL-1β treatment also increased the expression of Cathepsin B, which plays a pivotal role in lysosomal autophagy. This favoring of autophagy in cancer cells furthers cancer growth, as it favors clearance of metabolic waste from the cells. In contrast, IL-1RA at 1 µg/mL and Tolcapone at 20 µM inhibited autophagy, resulting in less proliferation and metastasis. These results clearly indicate the targeting of IL-1β and its signaling cascade as a promising therapeutic approach toward defeating glioblastoma growth.

### 2.7. IL-1β Down Regulates Apoptosis Resulting in U87 Cancer Cell Survival and Metastasis

Uncontrolled cell division and tumor metastasis are the two most important properties of cancer cells. This is the opposite of apoptosis, a cell death that hinders cancer cell proliferation. Suppression of apoptotic components like caspases in glioma cells results in malignant transformation and elevated metastasis. IL-1β treatment resulted in the down regulation of Caspase-8 and -3, thus favoring cancer cell proliferation (Figure 6). Conversely, IL-1RA and Tolcapone at 1 µg/mL and 20 µM/mL, respectively, increased the expression of Caspase-8 and Caspase 3 for induction of apoptosis and cell death of glioblastoma U87 cells. Tolcapone at 10uM was not effective in altering Caspase-8 levels but was effective at 20 μM.

## 3. Discussion

The evidence we provide here indicates a central role for neuroinflammation in glioblastoma, as we show, like Hurmath et al. 2014 [13], that elevated levels of IL-1β facilitate cancer cell proliferation and metastasis, which is reduced by treatment with IL-1RA. Here, we expand on these findings as we delineate the signaling mechanism whereby treatment with IL-1RA quells IL-1β’s dramatic effect toward glioblastoma pathogenesis and spread. In addition to IL-1RA, we report that the FDA-approved drug Tolcapone is as effective as IL-1RA in quelling this proliferation and metastasis. In this study, we discovered that IL-1β plays a profound role in facilitating the growth and metastatic potential of glioblastoma via IL-1β’s role in activating the TLR-MyD88-NFkB pathway. Our findings also highlight the role of IL-1β in activating autophagy mechanisms that are extremely beneficial in facilitating clearance of metabolic waste, thus allowing swift growth of cancer cells. From our previous studies, we reported the pathological role of IL-1β in Alzheimer’s and Parkinson’s diseases as well as other neuropathological conditions [5,18]. Rapid cell proliferation and tissue invasion are essential properties of cancer cells, which are dependent on sufficient autophagic clearance [19]. Evolutionarily, the immune system is highly regulated toward providing a check on important aspects of cellular metabolism and healthy cell proliferation. Cancer cells manipulate this immune system function to favor elevated cancer cell division as well as malignant transformation by activating signaling cascades required to suppress the host immune system. Cancer cells exploit several immunological processes such as targeting regulatory T cell functions, including their replication and the antigen presentation of such immune cells, thereby modifying the production of immune suppressive mediators, and simultaneously increasing the tolerance of cancer cells [20]. In response to uncontrolled cell proliferation, a neuroinflammatory environment is created—read as elevated levels of IL-1β—which, we elaborate here, favors cancer cell proliferation and spread. This proliferation and spread are further aided by IL-1β-induced autophagy. All these events clearly illustrate the seminal role of IL-1β in cancer cell proliferation and metastasis.

Toll-like receptors play an important role in the immune system in activation of the MyD88- and NFκB-signaling cascade for elevated production of inflammatory cytokines like IL-6 and IL-8, two cytokines that further activate immunological mechanisms to protect the cell from invading infection agents [21,22]. Lipopolysaccharides and cytokines like IL-1β and interferon-gamma [IFN-γ] activate the immune system to combat the threats to cellular systems [23], but in cancer cells, as we show here, IL-1β manipulates this evolutionarily protective property of the immune system toward malignant transformation of normal cells into cancer cells. Activation of either or both TLR2 and TLR4, and further downstream signaling via the MyD88 and NFκB pathway, has been shown in other cancerous tumors [23] as well as mycotic keratitis [20,21]. Over and above these roles in tumor progression, the TLR4/MyD88/NF-κB signaling pathway induces production of tumor necrosis factor-α, interleukin-6, and monocyte chemoattractant protein-1, all of which are associated with other conditions such as heart- and liver-related complications of Type-2 Diabetes Mellitus [24] and, interestingly, with Parkinson’s disease [18].

As we show in glioblastoma, IL-1β initiates a cascade of events similar to that in aggressive luminal-type breast cancer cell where mRNA and protein levels of MyD88 induce and activate NFĸB, which in turn activates IL-6 [25] for activation of Calcineurin [26]. This activation of MyD88 leads to the dephosphorylation of the nuclear factor kappa-B kinase subunit β [IKKβ], which cleaves the p65 peptide of the NFĸB complex, causing it to translocate to the nucleus, where it acts as a transcription factor for increases in the expression of IL-6 and Calcineurin B. IL-6 induces VEGF, which in turn increases angiogenesis and vascularization, necessary mechanisms for nourishment of tumor cells [25]. Calcineurin, an important phosphatase, plays a pivotal role in the dephosphorylation of cyclins and cyclin-dependent kinases, which themselves play a crucial role in the advancement of the cell cycle for cancer cell proliferation [27].

Autophagic clearance of metabolic waste is an especially important function in cancer cells, as it fosters clearance of metabolic waste from the cancer cells as they rapidly proliferate. Moreover, autophagy provides for recycling of their constituent metabolites for maintenance of cell survival and genetic stability, and even promotion of drug resistance and cancer cell survival, as it severely limits the efficacy of chemotherapeutic drugs [28]. Chloroquine, a well-known autophagy inhibitor, enhances chemosensitivity of brain tumors that have the B-RAF V600E mutation and in this way reduces drug resistance and enhances clinical outcomes [29]. Our present results clearly support a role for IL-1β in enhancing autophagy and thereby aiding cancer metastasis. IL-1RA and Tolcapone, like chloroquine, inhibit autophagy, suggesting the anticancer potential of these drugs. Interestingly, the use of small-molecule enhancement of autophagy through its clearance of metabolic waste has been suggested as a promising therapy in Alzheimer’s disease [30], consistent with the idea that drugs or small molecules targeting autophagy can act as a promising therapy in multiple conditions that are characterized by the overexpression of IL-1β.

Many existing therapies for cancer aim to enhance apoptosis by targeting different caspase pathways. While most of these do so indirectly, emerging insights into caspase functions and new therapeutic strategies based on directly targeting caspases are beginning to show some promise [31]. In the current study, our results show that IL-1β downregulates the expression of different caspases like caspase-8 and caspase-3, favoring cancer cell survival. In contrast, IL-1RA and the FDA-approved drug Tolcapone both induce apoptosis by upregulating the expression of caspase-8 and caspase-3, resulting in cancer cell death. FDA-approved anticancer drugs target cancer cell survival and proliferation by inducing apoptosis signaling pathways. Novel therapeutic compounds, such as Shepheridin, LY2181308 [an antisense oligonucleotide], EM-1421, and YM155, exhibit anti-tumor therapeutic efficacy in the clinical management of cancer patients by directly targeting the cancer cell protein “survivin” and/or inducing caspase expression and activity, ultimately resulting in apoptosis mediated via both intrinsic and extrinsic apoptotic pathways [32].

In conclusion, our data demonstrating the power of neuroinflammation [IL-1β] in promoting both cancer cell proliferation and metastasis is consistent with the idea that IL-1β is the primary target for eliminating cancer growth and metastasis in the brain, as we show here and logically propose in the body of this study. These findings suggest that therapies targeting IL-1β and its downstream events, as shown here, are promising in defeating the threat and incidence of cancers. Further, our results show that IL1-β promotion of the TLR-MYD88-NFkB-IL-6 pathway feeds back to further increase IL-1β itself, thereby creating a self-perpetuating cycle of cancer cell proliferation and metastasis., As we show here, this cycle can be challenged by IL-1RA and Tolcapone, and in this way, cancer growth and spread can be curbed.

## 4. Materials and Methods

### 4.1. MTT Assay

U87 cell lines were obtained from ATCC and tested negative for mycoplasma contamination. The effect of IL-1 β, IL-1RA, and Tolcapone on the viability of U87 glioblastoma cells was evaluated by MTT assay. Sub-confluent cells in DMEM complete medium [DMEM; Invitrogen/Life Technologies, Grand Island, NY, USA] supplemented with 10% *v*/*v* fetal bovine serum [FBS] were seeded at 5 × 10^4^ cells/mL per well in 96 well plates and incubated for 24 h. After incubation, cells were treated with various concentrations of Tolcapone [0.6–20 µg/mL] in DMEM-containing 0.5% FBS [incomplete DMEM] and incubated for 24 or 48 h. Untreated cells grown in incomplete media served as a negative control. Cells treated with 50 ng/mL of IL-1 β served as positive control. Cell viability was measured using the MTT assay as follows: 50 µL of MTT [5 mg/mL] prepared in incomplete DMEM in [1:1] was added to each well and further incubated for 2 h at 37 °C. After incubation, MTT was aspirated and insoluble crystals in each well were dissolved in DMSO [100 µL]. The absorbance was measured at 595 nm using a TECAN multimode microplate reader [M200 Pro]. The percentage of viable cell numbers was calculated with respect to controls [untreated], which were considered as 100%. Each experiment was performed in triplicate.

### 4.2. Immunofluorescence Microscopy

U87 cell lines were obtained from ATCC and tested negative for mycoplasma contamination. Cells grown in an 8-well chamber slide were fixed with 4% Paraformaldehyde, permeabilized with 0.2% Triton 100× in 1× PBS and blocked with 5% BSA-PBS-T [1× PBS with 0.1% Tween-20], followed by incubation with GFAP primary and secondary antibodies. GFAP antibody LB 509 [Santa Cruz, Cat# sc-58480] and secondary antibody for GFAP [Cat# GTX11267, Gene Tex] were used. After incubation with Alexa Fluor 488 [green] and 558 [red]-conjugated secondary antibodies, coverslips were mounted in DAPI [ThermoFisher, Mount Prospect, IL, USA]. Images were taken using an Olympus BX61-Regular Upright BF & Fluorescent/Reflect Microscope using a 20× objective. Images were analyzed with Image-J-win64, version 2.0, ROI Manager using the freehand selection tool. 

### 4.3. Clonogenic Assay

Glioblastoma U87 [0.2 × 10^6^] cells per well were seeded in 6-well plates and treated for 48 h with different concentrations of IL-1β, IL-IRA+IL-1β, or Tolcapone [IC_50_ concentration]. Cells were detached by trypsinization and seeded in complete medium [1 × 10^3^ cells/well] and incubated for 10 days for completion of 6–8 cycles by replacing the medium regularly at 48 h intervals. The colonies formed after 16 days were observed visually by fixing with glutaraldehyde [6.0% *v*/*v*] in PBS followed by staining with crystal violet [0.5% *w*/*v*]. The percentage of colony formation was compared between untreated cells, cells treated with IL-1β, IL-1RA, or Tolcapone.

### 4.4. Scratch Wound Healing Assay

The wound healing assay was performed to assess the metastatic potential of IL-1 β, IL-1RA+IL-1β, or Tolcapone on Glioblastoma U87 cells. U87 cells were seeded in 6-well plates [0.2 × 10^6^ cells/mL] and grown for 48 h to form a monolayer. A scratch wound was created in the monolayer using a sterile 200 μL pipette tip. Cells were washed with ice-cold PBS to remove scratch debris, followed by IL-1β treatment, IL-1RA+IL-1β, or Tolcapone in incomplete DMEM for different time intervals [24, 48, and 72 h]. The wound closure was observed and photographed using phase contrast microscopy.

### 4.5. RT-PCR Amplification

Total RNA was extracted from U87 cells treated with or without IL-1 β, IL-IRA+IL-1β, or Tolcapone for 48 h, using the Qiagen RNA extraction kit [RNeasy Plus Mini kit #74134, Qiagen, Venlo, The Netherlands], according to the manufacturer’s instructions. Quality and quantity of the extracted RNA was determined by Agilent bioanalyzer. RT reactions were performed on equal amounts of RNA using single-step RT-PCR reagents. The sequences of primers were made and purchased from IDT, as in the following Table 1.

RT-PCR conditions were as follows: holding stage [cDNA synthesis step] for 3 min at 50 °C, followed by a hold stage at 95 °C for 5 min and 40 cycles of 95 °C for 15 s, 60 °C for 30 s, followed by a hold stage at 40 °C for 1 min. All RT reactions were performed using 10 ng total RNA in 4 μL, at a concentration of 2.5 ng/μL. All mRNA levels mentioned in the manuscript were relative to the levels of 18S rRNA.

### 4.6. Western Blot Analysis

Western blot analysis was performed to determine the signaling proteins activated by IL-1β, with or without IL-IRA or Tolcapone treatment of U87 cells. Cells were treated with IL-1 β, IL-IRA+IL-1β, and Tolcapone [IC_50_] for 48 h, and then the cells were lysed using RIPA lysis buffer [120 mM NaCl, 1.0% Triton X-100, 20 mM Tris-HCl, pH 7.5, 100% glycerol, 2 mM EDTA, protease inhibitor cocktail], and the total protein was estimated. The protein was then electrophoresed on SDSPAGE for 2 h at 100 V on 4–20% gradient bis-tris acrylamide gels [BioRad Life Science, Hercules, CA, USA] and transferred to TRANS BLOT and blotted onto a PVDF membrane. The blots were blocked with blocking buffer [Pierce; Thermoscientific, Waltham, MA, USA] and incubated overnight at 4 °C with antibodies against signaling proteins, listed below. After overnight incubation with primary antibodies, the blots were washed 4 times for 5 min each. Membranes were incubated for 1 h at RT with HRP-conjugated secondary antibody—goat anti-rabbit and anti-mouse IgG, respectively [Cell signaling Technologies, 1:3000 dilution; Danvers, MA, USA], Cat #7074 and Cat #7076. GAPDH was used as an internal control. Blots were processed using an ECL chemiluminescence detection kit [Pierce]. Signals were recorded and documented using the Chemi-Doc Imaging system from Biorad and quantified using Image J software ImageJ [RRID:SCR_003070] [NIH Version 2.0].

The following commercially available antibodies were used: GAPDH mouse mAb from Santacruz [RRID:sc47724]; MyD88 [RRID:3699S] Rabbit mAb RRID: #12703 [Cell Signaling Technologies]; NF-κB p65 Rabbit mAb RRID:#50794 [catAb32536]; Rabbit mAb Cathepsin B [H190] [Antibody RRID:#3383 Cell signaling Technologies]; IL-6 [D5W4V] XP^®^ Rabbit mAb RRID:#12912 [Cell signaling Technologies]; 12912S Anti-Calcineurin B/PPP3R1 antibody EPR24992-18] RRID:ab303482 [Abcam]; Rabbit mAb for LC3B [cat RRID:#NB6001384] from Novus biologicals; LAMP-2; Caspase-3 Antibody RRID:#9662 [Cell signaling Technologies]; Caspase-8 [1C12] Mouse mAb RRID:#9746 [Cell Signaling Technologies]; Caspase-9 [9662S] Rabbit mAb RRID:#50794 [Cell Signaling Technologies].

### 4.7. Statistical Analysis

Sample size was based on estimations by power analysis with a level of significance of 0.05. The hypothesis was checked with the independent two-tailed *t*-test, which indicated a statistically significant difference in measurements between the group receiving placebo and the group receiving treatment; values were considered significantly different when the two-tailed *p* value was ≤0.05. (* indicates *p* < 0.05, ** indicates *p* < 0.005, *** indicates *p* < 0.0005). Results are expressed as mean SEM.

## Figures and Tables

**Figure 1 ijms-26-06893-f001:**
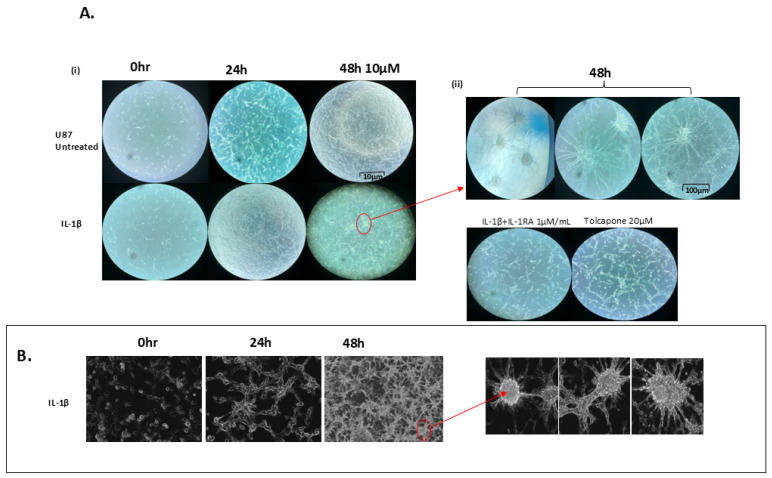
Morphological changes in IL-1β treated U87 cells. (**A**) (**i**) [24 h] Increased proliferation. (**ii**) [48 h] Formation of clone-like structures in IL-1β at 50 ng/mL treated U87 cells after 48 h treatment. (**B**) Microscopic images depicting IL-1β-driven formation of projections from U87 cells, forming connecting adjacent cells. IL-1RA and Tolcapone inhibited both proliferation and clonal formation as seen by fluorescence microscopy at 20× magnification.

**Figure 2 ijms-26-06893-f002:**
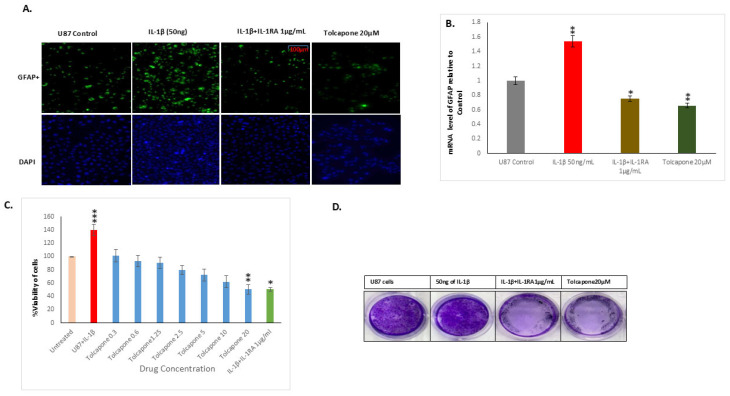
Immunofluorescence analysis for GFAP expression. (**A**). The intensity of the green color is directly proportional to the expression of GFAP protein in U87 cells, showing the extent to which IL-1RA and Tolcapone inhibited the expression of GFAP protein after 48 h. (**B**) IL-1β induction of GFAP mRNA levels. mRNA was extracted from IL-1β-, IL-1RA- and Tolcapone-treated U87 cells, and RT-PCR was performed. Quantification of the GFAP mRNA [relative to that for 18S]; values reflect the percentage of control as mean ± SEM. (**C**) Effects of IL-1β, IL-1RA, and Tolcapone on glioblastoma cancer cell growth as assessed by MTT assay. IL-1β at 50 ng/mL increased proliferation. Tolcapone at 20 µM strongly inhibited the 50% of the cancer cell growth. IL-1RA at 1 μg/mL strongly inhibited U87 cell growth at 48 h. Untreated cells were used as negative controls. Cells treated with IL-1β at 50 ng/mL were used as positive controls. (**D**) Clonogenic assay performed in six-well plates, with clones produced by U87 glioblastoma cancer cells. IL-1β at 50 ng/mL increases clonogenicity of human glioblastoma cancer cells: IL-1RA and Tolcapone reduced the number of colonies in U87 cells. Error bars represent SEM. * indicates *p* < 0.05, ** indicates *p* < 0.005 *** indicates *p* < 0.0005, via 1-tailed t test for an N of 3 biological repeats, represented as individual data points.

**Figure 3 ijms-26-06893-f003:**
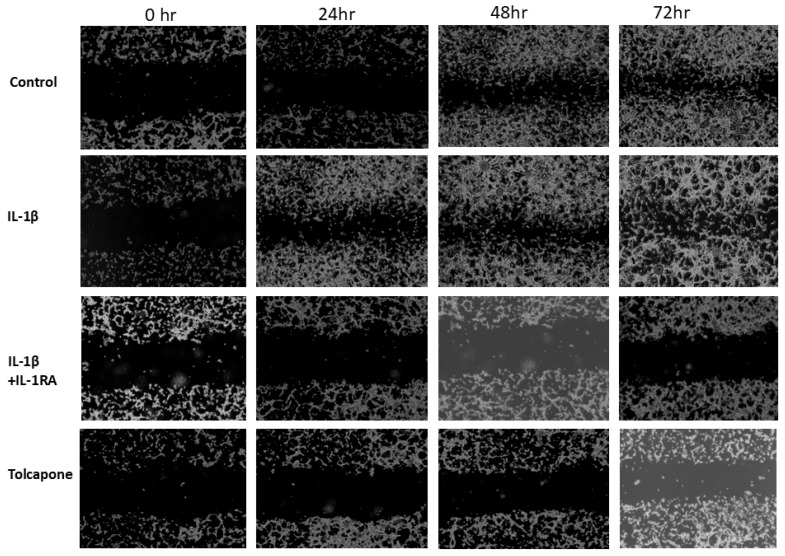
Scratch Wound Healing Assay for metastasis in U87 cells. Scratch wound healing assay of U87cells was performed to monitor the migration potential of IL-1β, IL-1RA and Tolcapone. IL-1β-induced increases in metastasis of U87 cells were inhibited by IL-1RA and Tolcapone. The wound closure was observed at indicated time points and photographed using a phase contrast microscope at 20× magnification.

**Figure 4 ijms-26-06893-f004:**
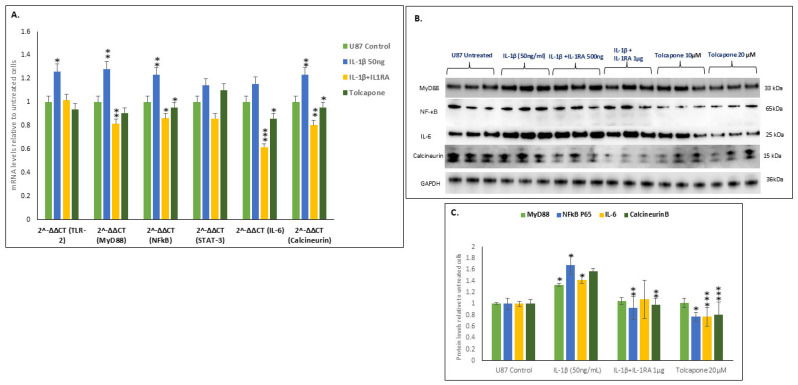
IL-1β increased the transcription and translation of the MyD88-NFĸB-IL-6-PPP3 pathway. (**A**). U87 cells were cultured for 24 h, and after reaching 70% confluency IL-1β, IL-1RA, and Tolcapone at the indicated concentrations were applied. After 48 h, mRNA levels of TLR2- MyD88-NFĸB-IL-6-PPP3 were determined by single step RT-PCR. Histogram shows means ± SEM. Significance of differences from untreated [*n* = 3] was determined by two-tailed *t*-tests within ANOVA [*p* < 0.01]. (**B**) Western-blot analysis of MyD88, NFĸB, IL-6 and Calcineurin proteins from IL-1β, IL-1RA and Tolcapone treated U87 cells at 48 h. Protein bands shown in (**B**) were derived from the same blot that was stripped and re-probed with different antibodies. (**C**) Histogram shows normalized band intensities from Western blots of from IL-1β-, IL-1RA-, and Tolcapone-treated U87 cells at 48 h. IL-1β at 50 ng/mL significantly increased protein levels of MyD88, NFĸB, IL-6, and Calcineurin in U87 cells. Values are mean  ±  SEM; significance between IL-1β-, IL-1RA-, and Tolcapone-treated cells and control cells was determined by 2-way ANOVA and the Bonferroni post hoc test, error bars represent SEM. * indicates *p* < 0.05, ** indicates *p* < 0.005, *** indicates *p* < 0.0005, via 1-tailed t test for an N of 3 biological repeats, represented as individual data points.

**Figure 5 ijms-26-06893-f005:**
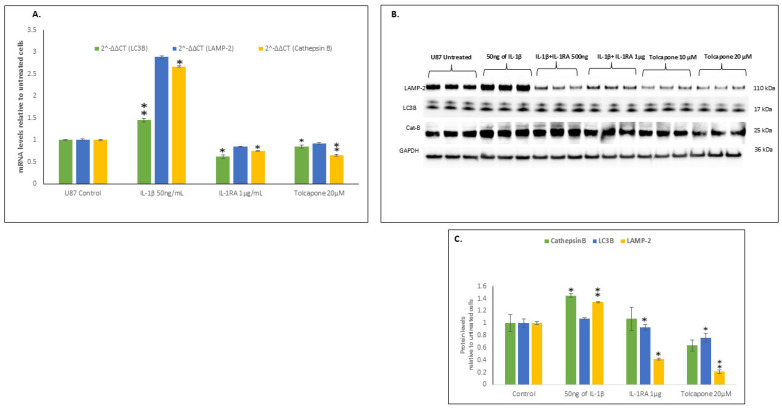
IL-1β increased the transcription and translation of autophagy mechanisms. (**A**) IL-1β upregulates autophagy genes in U87 cells. After 48 h, mRNA levels of autophagy gens, Cathepsin B, LC3B, and LAMP-2 were determined by single step RT-PCR. Histogram shows means ± SEM. Significance of differences from untreated [*n* = 3] was determined by two-tailed *t*-tests within ANOVA [*p* < 0.01]. (**B**) IL-1β upregulates autophagy proteins in U87 cells. Western-blot analysis of autophagy proteins, Cathepsin B, LC3B, and LAMP-2 from IL-1β-, IL-1RA-, and Tolcapone-treated U87 cells at 48 h. Protein bands shown in (**B**) were derived from the same blot that was stripped and re-probed with different antibodies. (**C**) Histogram shows normalized band intensities from Western blots of from IL-1β-, IL-1RA-, and Tolcapone-treated U87 cells at 48 h. IL-1β at 50 ng/mL significantly increased protein levels of Cathepsin B, LC3B, and LAMP-2 in U87 cells. Values are mean  ±  SEM; significance between IL-1β-, IL-1RA-, and Tolcapone-treated cells and control cells was determined by 2-way ANOVA and the Bonferroni post hoc test, error bars represent SEM. * indicates *p* < 0.05, ** indicates *p* < 0.005, via 1-tailed t test for an N of 3 biological repeats, represented as individual data points.

**Figure 6 ijms-26-06893-f006:**
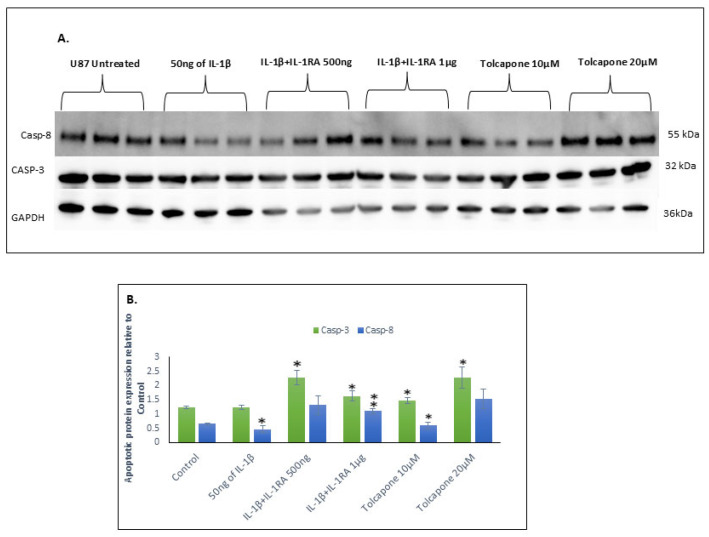
IL-1β down regulates apoptosis resulting in U87 cancer cell survival and metastasis. (**A**) Western-blot analysis of apoptotic proteins, Caspase-3 and -8 in IL-1β-, IL-1RA-, and Tolcapone-treated U87 cells at 48 h. Protein bands shown were derived from the same blot that was stripped and re-probed with different antibodies. (**B**) Histogram shows normalized band intensities from Western blots of from IL-1β-, IL-1RA-, and Tolcapone-treated U87 cells at 48 h. IL-1β at 50 ng/mL significantly decrease protein levels of Caspase-3 and 8; on the other hand, IL-1RA and Tolcapone increased the expression of Caspase-8 and Caspase 3 for induction of apoptosis and cell death of glioblastoma U87 cells. Values are mean  ±  SEM; significance between IL-1β-, IL-1RA-, and Tolcapone-treated cells and control cells was determined by 2-way ANOVA and the Bonferroni post hoc test, error bars represent SEM. * indicates *p* < 0.05, ** indicates *p* < 0.005, via 1-tailed t test for an N of 3 biological repeats, represented as individual data points.

**Table 1 ijms-26-06893-t001:** RT-PCR primers used.

Gene [Direction]	Sequence	Species
TLR-2	[F]5′-AAG GGC AGC TCA GGA TCT TT-3′ [R]5′-AGA CTG CCC AGG GAA GAA AA-3′	Human
MyD88	[F]5′-CCA GCA TTG AGG AGG ATT GC-3′ [R]5′-GCT CTG CTG TCC GTG GGA-3′	Human
NF-κB p65	[F]5′-AGA TAC CAC CAA GAC CCA CC-3′ [R]5′-CTG TCC CTG GTC CTG TGT AG-3′	Human
GFAP	[F]5′-GAGAGGGACAATCTGGCACA-3′ [R]5′-GGC TTC ATC TGC TTC CTG TCT-3′	Human
PPP3	[F]5′-AGA GGC AAA GGG TTT GGA TAG-3′ [R]5′-ATG TGC GGT GTT CAG AGA AT-3′	Human
IL-6	[F]5′-TGA AAG CAG CAA AGA GGC AC-3′ [R]5′-TCA CCA GGC AAG TCT CCT CAT-3′	Human
Cathepsin-B	[F]5′-TCT CTG ACC GGA TCT GCA TC-3′ [R]5′-TCA CAG GGA ATG GAG TA-3′	Human
LC3B	[F]5′-GTT ACG GAA AGC AGC AGT GTA-3′ [R]5′-CAG AAG GGA GTG TGT CTG AAT G-3′	Human
Lamp2	[F]5′-GAA ATG CCA GTG TGT CCT AGA-3′ [R]5′-TCC CAA AGT GCT GGG ATT AC-3′	Human
18S	[F]5′-TTC GGA CGT CTG CCC TAT CAA-3′ [R]5′-ATG GTA GGC ACG GCG ACT A-3′	Human

## Data Availability

Data sharing is not applicable to this article as no datasets were generated or analyzed during the current study.

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
