# Peer review of "The Seminal Role of the Proinflammatory Cytokine IL-1β and Its Signaling Cascade in Glioblastoma Pathogenesis and the Therapeutic Effect of Interleukin-1β Receptor Antagonist (IL-1RA) and Tolcapone [Author-notes fn1-ijms-26-06893]"

_ijms, 2025, doi:10.3390/ijms26146893_

Round 1

Reviewer 1 Report

Comments and Suggestions for Authors

This article is devoted to the study of the role of IL-1β in the activation of the signaling cascade that leads to proliferation and metastasis in the pathogenesis of glioblastoma cancer, as well as the therapeutic role of IL-1 receptor antagonist and tolcapone against the untoward aspects of tumor pathogenesis. In the introduction, the authors introduce us to the research problematic, set adequate goals and corresponding tasks. The obtained results are well discussed both in the context of this work and compared with the work of other laboratories.

Unfortunately, there are a number of comments on the work, both regarding the text and the quality of the illustrations, as well as the statistical analysis of the data obtained.

L. 16. it is better to insert treatment or cultivation with IL-1β at...

L.22 missed of - "levels of cathepsin"...

L. 24 it is better to use phrase - pro-apoptotic proteins such as...

L. 26 from here onwards, review the text for missing spaces

L.55-60 -  this paragraph needs to be more connected to the previous one... a bit out of context of the work...

L. 57 It should be written - mutations in such and such gene, not "Mutations in calcineurin, but "mutations in the gene PPP3CC (https://www.genecards.org/cgi-bin/carddisp.pl?gene=PPP3CC)

L.70 insert a few sentences about what kind of drug (Tolcapone) it is - for example, it is used against Parkinson's disease.

L. 72 Please check - U87 MG

L. 77 - in the abstract there is also about caspase 8, here only about the third...

Figures 1 -6 - here and below remove the title of the drawing from the picture. The title of the picture goes below its illustration.

Figure 1A. a bit of a strange name, avoid such phrases - there are statistics that will calculate for you how "dramatic" it is. And since the resolution also suffers, it's not quite Stanislavsky method yet...

Figure 1A ii - what is this, there are no captions, the drawing (Figure 1A) as a whole is poorly grouped

Figure 1B -  the picture on the right - what is that? How many hours?

Figure 1 - How many fields did you watch? Did you or didn't you calculate statistically in the ImageJ (for example)?

Figure 2 A and D - poor resolution. Microscopic photographs must have a scale.

Figure 2 A  - Have you tried cytofluorometry? Then you would have a quantitative analysis. This remark is advisory in nature.

Figure 2 С -  Judging by the picture, you practically do not achieve IC50 for Tolcapone, although in the text  (L.345) you write that there is data, how was it calculated then?

Figure 2D - Please see the manuscript how to format the results of the clonogenic test, you have only quality images. Franken, N., Rodermond, H., Stap, J. et al. Clonogenic assay of cells in vitro. Nat Protoc 1, 2315–2319 (2006). https://doi.org/10.1038/nprot.2006.339

L.105-111 It is precisely in the text that it is necessary to indicate IC50, and not other inhibitory concentrations.

Figure 2 - The methodological part should be moved to the appropriate section, leaving only the title and notes regarding statistics. Between pictures 2 and 3, you need to place text dedicated to wound assay.

Figure 3 - There are programs that calculate the results of wound assays. These pictures can also be provided, but it is necessary to conduct a quantitative analysis of them.

Figure 4A  - Make all your bar charts in the same color scheme and style - yours is very colorful.

Figure 4С -   "IL-1β + IL-1RA 1 mkg"  bars - It is doubtful that there is such a level of reliability - with such SD, it is possible if you have a very large sample power. Please check. In the "Tolcapone 20mkg" group MyD88 does not decrease, where do the reliability values ​​come from?

Figure 5 A, C - Make the asterisks bigger, they are not visible at all.

Figure 6 - Why does the level of caspase-8 protein decrease in the group "Tolcapone 10 mkg"?

L. 316 -Please name the subsection simply - MTT assay

L. 426 An analysis of the Reference Literature revealed many old manuscripts.

In general, with a more scrupulous and accurate presentation of the results obtained, rechecking the statistical analysis, the article can be published in this highly rated journal, since it really is of scientific interest. One of the disadvantages of the article is also the performance of all analyses on one cell line, however, if we consider the article as a proof of principle, then it quite scientifically and convincingly proves the authors' hypotheses.

Author Response

Dear Reviewer,

We would like to express our sincere thanks for your time and consideration for reviewing our submission, entitled ‘The seminal role of the proinflammatory cytokine IL-1β and its signaling cascade in glioblastoma pathogenesis and the therapeutic effect of Interleukin-1β Receptor Antagonist (IL-1RA) and Tolcapone. We are especially appreciative of the laudatory comments and wish to reply to the reviewers.

We incorporated all the necessary changes in the revised manuscript as suggested. This reviewer raises an interesting point, which we address in the following ways.

  1. 16. it is better to insert treatment or cultivation with IL-1β at...

Response- As per the reviewer comments we corrected the sentence.

L.22 missed of - "levels of cathepsin"...

Response- As per the reviewer comments we corrected the sentence.

  1. 24 it is better to use phrase - pro-apoptotic proteins such as...

Response- As per the reviewer comments we corrected the sentence.

  1. 26 from here onwards, review the text for missing spaces

Response- As per the reviewer comments we corrected the sentence.

L.55-60 -  this paragraph needs to be more connected to the previous one... a bit out of context of the work...

Response- As per the reviewer comments we corrected the sentence.

  1. 57 It should be written - mutations in such and such gene, not "Mutations in calcineurin, but "mutations in the gene PPP3CC (https://www.genecards.org/cgi-bin/carddisp.pl?gene=PPP3CC)

Response- As per the reviewer comments we corrected the sentence.

L.70 insert a few sentences about what kind of drug (Tolcapone) it is - for example, it is used against Parkinson's disease.

Response- As per the reviewer comments we included the sentence regarding Tolcapone

  1. 72 Please check - U87 MG

Response- We apologize for the typo and as per the reviewer comments we changed to MG

  1. 77 - in the abstract there is also about caspase 8, here only about the third...

Response- As per the reviewer comments we included caspase 8 in the sentence.

Figures 1 -6 - here and below remove the title of the drawing from the picture. The title of the picture goes below its illustration.

Response- As per the reviewer’s comments the tiles of the pictures have been deleted.

Figure 1A. a bit of a strange name, avoid such phrases - there are statistics that will calculate for you how "dramatic" it is. And since the resolution also suffers, it's not quite Stanislavsky method yet...

Response. We welcome the reviewer’s response and as the figure states the morphological changes after treatment we decided to go with the title “Morphological changes in IL-1β treated U87 cells”

Figure 1A ii - what is this, there are no captions, the drawing (Figure 1A) as a whole is poorly grouped

Response- Thank you for the comment and as per the reviewer’s comments we corrected the figure 1A ii. and included in the revised manuscript. Kindly note the figure caption where we clearly illustrated Fig 1 A. [i] and [ii] in the legend.

Figure 1B -  the picture on the right - what is that? How many hours?

Response- Thank you for the comment and as per the reviewer comments we corrected the figure 1A ii. and included the time point in the revised manuscript.

Figure 1 - How many fields did you watch? Did you or didn't you calculate statistically in the ImageJ (for example)?

Response- We watched under two fields, and we didn’t calculate statistically using ImageJ as it’s just the morphological changes depicting the increased proliferation which was quantified by MTT and presented in the Figure 2 C.

Figure 2 A and D - poor resolution. Microscopic photographs must have a scale.

Response- Microscopic photographs with scales have been added in the revised manuscript.

Figure 2 A  - Have you tried cytofluorometry? Then you would have a quantitative analysis. This remark is advisory in nature.

Response- Thank you for the reviewer’s advice and we don’t have cytofluorometry for quantitative analysis.  

Figure 2 С -  Judging by the picture, you practically do not achieve IC50 for Tolcapone, although in the text (L.345) you write that there is data, how was it calculated then?

Response- In the MTT assay Tolcapone at 20 µM inhibited 50% of the cell viability and we considered that concentration (20 µM) as a IC50 by MTT Assay.

Figure 2D - Please see the manuscript how to format the results of the clonogenic test, you have only quality images. Franken, N., Rodermond, H., Stap, J. et al. Clonogenic assay of cells in vitro. Nat Protoc 1, 2315–2319 (2006). https://doi.org/10.1038/nprot.2006.339

Response- Thank you for the reviewer’s advice and Figure 2 and its legend has been corrected and included in the revised manuscript.

L.105-111 It is precisely in the text that it is necessary to indicate IC50, and not other inhibitory concentrations.

Response- Appropriate changes have been made and included in the revised manuscript.

Figure 2 - The methodological part should be moved to the appropriate section, leaving only the title and notes regarding statistics. Between pictures 2 and 3, you need to place text dedicated to wound assay.

Response- Appropriate changes have been made and included in the revised manuscript.

Figure 3 - There are programs that calculate the results of wound assays. These pictures can also be provided, but it is necessary to conduct a quantitative analysis of them.

Response- This reviewer raises an interesting point, which we address in the following ways. Wound healing assay and clonogenic assay were already reported by Hurmath et,al 2013 and here we used our experiments to show it as supportive data and it was not a focus of the paper. The focus of the paper was to show the signaling cascade induced by IL-1β and how it could be helpful in the metastasis of the Glioblastoma pathogenesis.

Figure 4A  - Make all your bar charts in the same color scheme and style - yours is very colorful.

Response- As per the reviewer’s comments all the bar diagrams have been changed to same color in revised manuscript.

Figure 4С -   "IL-1β + IL-1RA 1 mkg"  bars - It is doubtful that there is such a level of reliability - with such SD, it is possible if you have a very large sample power. Please check. In the "Tolcapone 20mkg" group MyD88 does not decrease, where do the reliability values ​​come from?

Response- We sincerely apologize for the mistakes we made in this figure. Thank you for the reviewers’ comment on this aspect and we corrected the image and indeed Tolcapone did not decrease MyD88 levels and appropriate changes have been made and included in the revised figure and the manuscript as well.

Figure 5 A, C - Make the asterisks bigger, they are not visible at all.

Response- Appropriate changes have been made and included in the revised manuscript.

Figure 6 - Why does the level of caspase-8 protein decrease in the group "Tolcapone 10 mkg"?

Response- This reviewer raises an interesting point, Tolcapone was not effective in altering Caspase-8 levels at 10uM but was effective at 20uM and the same was mentioned in the revised manuscript.

  1. 316 -Please name the subsection simply - MTT assay

Response- Appropriate changes have been made and included in the revised manuscript.

  1. 426 An analysis of the Reference Literature revealed many old manuscripts.

Response- Appropriate changes have been made and included in the revised manuscript.

Once again, Thank you very much for your time and consideration.

Sincerely yours,

  1. Sue T. Griffin, PhD, Founder, Journal of Neuroinflammation
    Professor and Vice Chairman for Research
    Donald W. Reynolds Department of Geriatrics

Donald W. Reynolds Institute on Aging

University of Arkansas for Medical Sciences

Research Scientist Geriatric Research Education Clinical Center

Central Arkansas Veterans Health Care System

Little Rock, Arkansas 72205

Mobile: 501-766-6530 (griffinsuet@uams.edu)

Reviewer 2 Report

Comments and Suggestions for Authors

The manuscript is devoted to investigating the effect of interleukin-1 receptor antagonist and Tolcapone on glioblastoma cell proliferation. The study is accurately described and well presented. The methods are described in detail and are adequate to address the questions raised in the study. However, there are still points in the initial study design that should be highlighted prior to publishing.
First of all, the U87 cell line is well-established, and the role of interleukin-1b in its cell cycle was investigated years ago. Apart from the mentioned study by Hurmath et al., several other papers are reporting not only the increased proliferation of U87 cells after exposure to interleukin-1β, but also the interleukin-1β-induced apoptosis of U87 cells in a hypoxic environment (which is also a physiological condition for glioblastoma cells)- Sun W. et al., Cell Death Dis., 2014, doi: 10.1038/cddis.2013.562. Thus, the reason for selecting only this cell line should be clearly explained, as well as a comparison with the results obtained for other cell lines (at least those reported in the literature) should be provided and discussed. Additionally, it is unclear in what exact condition the cells were incubated, either with or without treating agents.
Second, it is clear why to assess the effect of  interleukin-1b Receptor Antagonist, but what was the reason for assessing Tocapone in this study? How is it connected with interleukin-1b? Are there biological reasons to assess both agents (and only them) in this particular study? What is similar (or different) in the results (apart from just cell proliferation and survival analysis) from the molecular viewpoint?

Additionally, as a minor suggestion, please replace square brackets with round ones.

Author Response

Dear Reviewer,

We would like to express our sincere thanks for your time and consideration for reviewing our submission, entitled ‘The seminal role of the proinflammatory cytokine IL-1β and its signaling cascade in glioblastoma pathogenesis and the therapeutic effect of Interleukin-1β Receptor Antagonist (IL-1RA) and Tolcapone. We are especially appreciative of the laudatory comments and wish to reply to the reviewers.

We incorporated all the necessary changes in the revised manuscript as suggested. This reviewer raises an interesting point, which we address in the following ways.

Reviewer comment-1, the U87 cell line is well-established, and the role of interleukin-1b in its cell cycle was investigated years ago. Apart from the mentioned study by Hurmath et al., several other papers are reporting not only the increased proliferation of U87 cells after exposure to interleukin-1β, but also the interleukin-1β-induced apoptosis of U87 cells in a hypoxic environment (which is also a physiological condition for glioblastoma cells)- Sun W. et al., Cell Death Dis., 2014, doi: 10.1038/cddis.2013.562. Thus, the reason for selecting only this cell line should be clearly explained, as well as a comparison with the results obtained for other cell lines (at least those reported in the literature) should be provided and discussed. Additionally, it is unclear in what exact condition the cells were incubated, either with or without treating agents.

Response-
This reviewer raises an interesting point, which we address in the following ways. As previously reported by Hurmath et,al 2013 showed the role of IL-1β is an important factor that can influence glioblastoma cell migration, invasion and proliferation likely via IL-1 receptor activation. Here our experiments show the molecular signaling mechanism underlying and the therapeutic effects of IL-1RA and the FDA approved drug Tolcapone as a possible therapeutic intervention. 
Reviewer comment-2, it is clear why to assess the effect of interleukin-1b Receptor Antagonist, but what was the reason for assessing Tolcapone in this study?

How is it connected with interleukin-1b? Are there biological reasons to assess both agents (and only them) in this study? What is similar (or different) in the results (apart from just cell proliferation and survival analysis) from the molecular viewpoint?

Response-
This reviewer raises an interesting point, which we address in the following ways. Tolcapone is an FDA approved drug currently under phase trials in the treatment for Parkinsons disease in the adults. Tolcapone has been shown a promising anti-cancer effect in Neuroblastoma cancer cells by inducing apoptosis. So, using FDA approved drug with known mechanism of action with other cancer cells prompted us to explore its anti-cancer potential in Glioblastoma cancer cells. As it is a preliminary data, more detailed study with animal models will give us more in-depth of these new therapeutic interventions.

Once again, Thank you very much for your time and consideration.

Sincerely yours,

  1. Sue T. Griffin, PhD, Founder, Journal of Neuroinflammation
    Professor and Vice Chairman for Research
    Donald W. Reynolds Department of Geriatrics

Donald W. Reynolds Institute on Aging

University of Arkansas for Medical Sciences

Research Scientist Geriatric Research Education Clinical Center

Central Arkansas Veterans Health Care System

Little Rock, Arkansas 72205

Mobile: 501-766-6530 (griffinsuet@uams.edu)

Round 2

Reviewer 2 Report

Comments and Suggestions for Authors

Thank you for the detailed response, however, I suggest to add the following key points according to the response provided:

  1. Explicitly state the study limitation regarding the only U87 model used;
  2. Include (in the Introduction section) the detailed motivation to study Tolcapone in this work with appropriate references to previous works regarding its anti-cancer effect.

Author Response

Dear Reviewer,

We would like to express our sincere thanks for your valuable suggestions in response to our submission paper entitled, “The seminal role of the proinflammatory cytokine IL-1β and its signaling cascade in glioblastoma pathogenesis and the therapeutic effect of Interleukin-1β Receptor Antagonist (IL-1RA) and Tolcapone” for publication in International Journal of Molecular Sciences under a Special Issue entitled "The Impact of the Immune System on the Tumor Microenvironment: Relevance to Glioma Development."

Here we are submitting a revised manuscript after considering your valuable suggestions. In the revised manuscript we incorporated all the suggestions and are addressed as follows.

  1. Explicitly state the study limitation regarding the only U87 model used;

Response. We sincerely appreciate the reviewer’s suggestion, and we included the appropriate response as follows

Hurmath et, al 2014 reported that overexpression of IL-1β in human glioblastoma cells promoted the proliferation and migration of U87MG and U251MG cells, an event that was blocked by treating the cells with IL-1Receptor antagonist [IL-1RA] [13], resulting in inhibition of migration, invasion, and proliferative effects, indicating that the effects were mediated by IL-1β and its receptor. IL-1 β significantly increased proliferation of U87MG glioma cells compared to control and interestingly there was no proliferation in U251MG glioma cells compared to control. The U87 cell line is used to study cancer growth and development, underlying molecular mechanisms, signaling pathways, and tumor microenvironment U-87MG cell lines are well studied glioblastoma cell lines for drug screening and testing purposes, allowing researchers to identify new potential anticancer drugs and assess their efficacy and toxicity. This prompted us to use U87MG cell line as a model cell line to understand the signaling mechanisms and to explore the therapeutic applications of IL-1RA and the FDA approved drug Tolcapone.

  1. Include (in the Introduction section) the detailed motivation to study Tolcapone in this work with appropriate references to previous works regarding its anti-cancer effect.

Tolcapone, a drug commonly used in the treatment of Parkinson's disease, is a po-tent inhibitor of Catechol‐O‐methyltransferase (COMT) and is permeable across the blood–brain barrier [14].  Previous studies indicate that Tolcapone increases the bioa-vailability of dopamine in cells. Tolcapone is FDA approved in adult patients for the treatment of Parkinson's disease (PD) as an adjunct therapy with levodopa, which is a dopamine precursor [15]. Interestingly, Tolcapone has shown a promising anticancer potential with neuroblastoma (NB) cells in preclinical models by inhibiting COMT. Treating four established NB cells lines (SMS‐KCNR, SH‐SY5Y, BE (2)‐C, CHLA‐90) and two primary NB cell lines with Tolcapone for 48 h decreased cell viability in a dose‐dependent manner, and induces cell death by inducing caspase‐3‐mediated apoptosis in neuroblastoma cell lines [16]. This prompted us to study the anti-cancer potential of Tolcapone against the glioblastoma pathogenesis using U87 MG as a model cell line. 

Once again thank you for your time and consideration.

Sincerely yours,

  1. Sue T. Griffin, PhD, Founder, Journal of Neuroinflammation

Professor and Vice Chairman for Research

Donald W. Reynolds Department of Geriatrics

Donald W. Reynolds Institute on Aging

University of Arkansas for Medical Sciences

Research Scientist Geriatric Research Education Clinical Center

Central Arkansas Veterans Health Care System

Little Rock, Arkansas 72205

Mobile: 501-766-6530 (griffinsuet@uams.edu)
